# DiSTAR: Diffusion over a Scalable Token Autoregressive Representation for Speech Generation

## Abstract

Recent attempts to interleave autoregressive (AR) sketchers with diffusion-based refiners over continuous speech representations have shown promise, but they remain brittle under distribution shift and offer limited levers for controllability. We introduce DiSTAR, a zero-shot text-to-speech framework that operates entirely in a discrete residual vector quantization (RVQ) code space and tightly couples an AR language model with a masked diffusion model, without forced alignment or a duration predictor. Concretely, DiSTAR drafts block-level RVQ tokens with an AR language model and then performs parallel masked-diffusion infilling conditioned on the draft to complete the next block, yielding long-form synthesis with blockwise parallelism while mitigating classic AR exposure bias. The discrete code space affords explicit control at inference: DiSTAR produces high-quality audio under both greedy and sample-based decoding using classifier-free guidance, supports trade-offs between robustness and diversity, and enables variable bit-rate and controllable computation via RVQ layer pruning at test time. Extensive experiments and ablations demonstrate that DiSTAR surpasses state-of-the-art zero-shot TTS systems in robustness, naturalness, and speaker/style consistency, while maintaining rich output diversity. Audio samples are provided on
`https://anonymous.4open.science/w/DiSTAR_demo`.

## 1 Introduction

Zero-shot text-to-speech (TTS) aims to synthesize natural, intelligible speech for an unseen speaker, matching voice timbre and style from only a brief prompt while remaining robust over long passages (Anastassiou et al., 2024; Du et al., 2024b; Chen et al., 2025b). This capability is central to open-domain narration, content creation, and conversational agents (Ji et al., 2024; Nguyen et al., 2023).

A first mainstream route models speech as a sequence of discrete tokens from a single codebook (e.g., from a neural audio codec) and applies an autoregressive language model (AR) to predict the next token, followed by a vocoder/codec decoder to reconstruct the waveform Anastassiou et al. (2024); Shen et al. (2023). This track inherits mature AR modeling and decoding strategies from the Natural Language Processing community: discrete `[EOS]` tokens enable clear termination; maximum-likelihood training is relatively stable compared with Mean Squared/Absolute Error Loss; and decoding hyper-parameters (temperature, top-$p$/top-$k$) and perplexity provide interpretable control at inference. However, codec rate sequences are long; exposure bias compounds in thousands of steps; and purely AR decoders struggle with long-range consistency (speaker/style drift) and throughput Song et al. (2024). Moreover, when a single discrete layer runs at low bitrate or with limited expressivity, reconstructions tend to lose fidelity and fine detail.

A second line of work models continuous speech representations (e.g., mel spectrograms or latents from pre-trained audio variational autoencoder (Kingma et al., 2019) or self-supervised learning encoders) and generates in the continuous space via diffusion or continuous-domain AR before vocoding to waveform (Chen et al., 2025b; Eskimez et al., 2024). Recent work interleaves AR drafting with *next-patch diffusion* over continuous latents, balancing quality and compute, and achieving impressive zero-shot cloning and cross-speaker generalization (Jia et al., 2025). Yet continuous latents introduce practical fragilities: modeling high-dimensional, information-dense features complicates

optimization and convergence; systems are sensitive to domain shift; many rely on explicit duration predictors; and pipelines that add reinforcement learning (RL) or intricate aligners raise engineering and tuning burden.

A third, increasingly influential direction embraces multi-codebook discrete representations, typically residual vector quantization (RVQ) (Lee et al., 2022), where several codebooks per frame progressively capture detail (Chen et al., 2024; Xiaomi, 2025). RVQ confers two attractive properties: (i) at a sufficient aggregate bitrate it can reconstruct high-fidelity audio; and (ii) by remaining discrete, it preserves the stability and interpretability of LM-style training and decoding. However, RVQ introduces a second dependency axis beyond temporal order: intra-frame depth – the strong correlation among codebooks at the same time frame – is critical for quality. Effective RVQ TTS systems must therefore model time and depth jointly. Previous explorations, such as including flattening codebooksBorsos et al. (2022), semantic-to-acoustic hierarchiesChen et al. (2025a), RQ-TransformerYang et al. (2023), and delay-pattern schedulingCopet et al. (2023), partially address this coupling but still trade off inference parallelism, long-range consistency, and train/infer efficiency, leaving RVQ's full potential under-exploited. This raises a central question: Can we architect a generator that natively models RVQ's joint time-depth structure, achieving high quality at reasonable compute?

To address this challenge, we introduce DISTAR (**DI**ffusion over a **S**calable **T**oken **A**uto**R**egressive Representation for Speech Generation), a zero-shot TTS framework that operates entirely in the RVQ discrete code space and tightly couples an AR language model with a masked diffusion transformer. DISTAR adopts the patch-wise factorization strategy popularized in next-patch systems Jia et al. (2025): it aggregates RVQ tokens into patches. A causal LM drafts the next patch by predicting a compact hidden sketch that captures coarse temporal evolution, after which a discrete masked diffusion Transformer performs parallel infilling within the patch. Inspired by LLaDA Nie et al. (2025), the masked diffusion component operates not as a continuous denoiser but as a iterative discrete demasking process over masked positions, thereby resolving multi-codebook (depth) dependencies and supporting efficient parallel synthesis in the RVQ code space.

The design gives two main advantages. Compared with continuous next-patch diffusion, DISTAR avoids optimization issues in high-dimensional continuous latents while retaining patch-level parallelism. Compared with single-codebook AR, it models the intra-frame multi-codebook coupling inside each patch, improving coherence and allowing depthwise parallel refinement, which reduces exposure-bias effects without sacrificing the robustness of discrete training.

Several design choices in DISTAR yield practical advantages. First, the fully discrete setting preserves an [EOS] token for the immediate termination of patch-level generation, eliminating auxiliary duration predictors or stop heads Jia et al. (2025); Chen et al. (2025b); Shen et al. (2018). Second, sharing the RVQ code space between the AR sketcher and the masked diffusion refiner allows end-to-end optimization and reduces inter-module mismatch relative to cascaded pipelines Wang et al. (2024); Du et al. (2024b). Third, decoding achieves robust quality under purely greedy settings, while temperature-based sampling extends to RVQ-level and joint layer–time strategies, enabling fine-grained control over the diversity–determinism trade-off. Fourth, pruning the upper RVQ layers at inference controls computation and bitrate to match bandwidth/latency constraints without retraining.

On standard zero-shot TTS benchmarks, DISTAR demonstrates strong robustness, speaker similarity, and naturalness, while maintaining the inference cost close to its continuous counterpart DiTAR and using fewer/comparable parameters than competitive baselines. Audio samples are available on the demo page.

In summary, our contributions to the community include the following:

- We introduce DISTAR, a zero-shot TTS framework that couples an autoregressive drafter with masked diffusion entirely in the discrete RVQ domain, achieving patch-level parallelism and joint modeling of RVQ layer–time dependencies.

- We develop an RVQ-specific sampling method that boosts quality and stability; support for diverse decoding strategies, and on-the-fly bitrate/compute control without retraining.

- In standard zero-shot benchmarks, DISTAR provides state-of-the-art robustness, speaker similarity, and naturalness with comparable or lower computational cost.

## 2 RELATED WORKS

### 2.1 ZERO-SHOT TEXT-TO-SPEECH

Zero-shot TTS work broadly bifurcates by the target representation. Continuous-latent approaches predict high-information features (mel or codec latents) with diffusion/flow to boost long-range consistency and robust cloning. (Shen et al., 2023; Ju et al., 2024) scale latent and factorized diffusion with speech prompting; (Le et al., 2024) casts text-guided speech infilling as flow matching; (Eskimez et al., 2024) and (Chen et al., 2025b) report strong results via continuous flow matching; (Yang et al., 2025) simplify training with scalar-latent codecs and Transformer diffusion; (Li et al., 2024) improves time-varying style control; and recent (Lee et al., 2024) and (Liu et al., 2024) explore DiT-based and autoregressive diffusion decoders, while (Jia et al., 2025; Sun et al., 2024) place diffusion heads inside causal stacks to retain AR-like controllability. In contrast, discrete-token pipelines discretize speech and leverage AR LMs for stronger in-context prompting and explicit decoding control. (Chen et al., 2025a) set the Nerual codec LM recipe, with (Chen et al., 2024; Song et al., 2025b) improving robustness and human parity; token-infilling AR models such as (Kharitonov et al., 2023) and (Peng et al., 2024) advanced editing and wild-data zero-shot; (Wang et al., 2024) uses masked generative codec modeling; and large multilingual systems such as(Anastassiou et al., 2024; Du et al., 2024a; Casanova et al., 2024), broaden coverage and controllability.

### 2.2 MASK DIFFUSION MODELS

Masked diffusion extends discrete-state diffusion to language by formalizing token corruption with structured transition kernels, notably D3PM and multinomial diffusion, which introduced absorbing masks and principled categorical noising (Austin et al., 2021; Hoogeboom et al., 2021). Recent theory shows that masked-diffusion training objectives reduce to mixtures of masked-LM losses and can be reparameterized to connect tightly with any-order autoregression; this yields simpler sampling and clarifies likelihood bounds (Ou et al., 2024; Shih et al., 2022). Scaling experiments train masked-diffusion LMs from scratch at LLM scale (e.g., LLaDA), demonstrating competitive perplexity and strong instruction-following after SFT (Nie et al., 2025). Beyond pure masking, generalized interpolating discrete diffusion mixes masking with uniform noise, enabling self-correction and compute-matched gains (von Rütte et al., 2025). Parallel advances include simplified/standardized masked diffusion parameterizations, reparameterized discrete diffusion routes and denoisers, and analyses that decouple paradigm (MDM vs. AR) from architecture (Shi et al., 2024; Zheng et al., 2023). Task-level studies further adapt discrete diffusion to conditional long-text generation with improved efficiency (Dat et al., 2024). Collectively, these results position masked discrete diffusion as a competitive LM family with any-order decoding, efficient parallelism, and growing theoretical clarity.

## 3 DISTAR

### 3.1 OVERVIEW

Figure 1 summarizes DISTAR, an autoregressive architecture that advances patch by patch while remaining entirely within a discrete residual vector-quantized (RVQ) code domain.

#### 3.1.1 FORMULATION

We recast long sequences of discrete speech codes into a tiled layout of patches, akin to the DiTAR (Jia et al., 2025) paradigm. An autoregressive (AR) Transformer governs dependencies across patches, while a masked diffusion Transformer completes the contents within each patch in parallel. In practice, generation advances along the patch index autoregressively, with the next patch produced via conditional masked diffusion. Let $\mathbf{C} = [\mathbf{c}_0, \mathbf{c}_1, \ldots, \mathbf{c}_{L-1}] \in \mathbb{Z}^{L \times J}$ denote the sequence of RVQ codes from $J$ quantizers, where $L$ is the RVQ code sequence length. And let $\mathbf{X}$ be the input text. We estimate the DISTAR parameters $\theta$ by conditional maximum likelihood over the patch-grouped codes. By the chain rule, the model factorizes as

$$p_\theta(\mathbf{C} \mid \mathbf{X}) = \prod_{i=0}^{L-1} p_\theta(\mathbf{c}_i \mid \mathbf{c}_{<i}, \mathbf{X}), \tag{1}$$

where $\mathbf{c}_{<i} = [\mathbf{c}_0, \ldots, \mathbf{c}_{i-1}]$ collects all previously generated codes. Training minimizes the negative log-likelihood, while inference realizes the autoregressive step at the patch level and resolves intra-patch tokens via masked diffusion in an iterative parallel decoding pass.

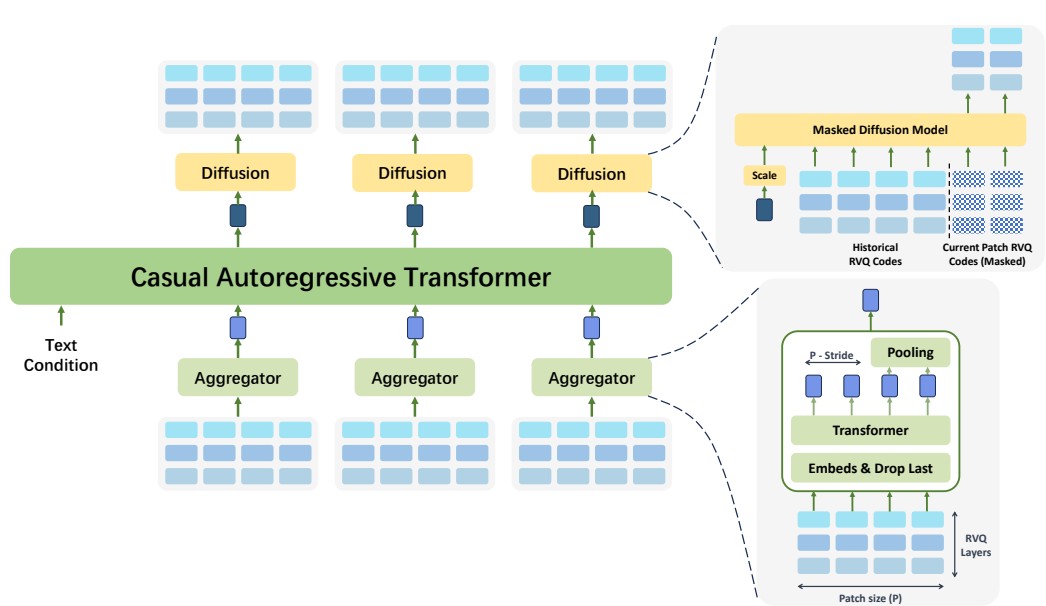

Figure 1: An overview of the proposed DiSTAR framework. It consists of an aggregator for RVQ code input, a causal language model backbone with text prompts, and a masked diffusion decoder, predicting patches of tokens purely on the discrete RVQ space.

**Patchified view of the code stream.** Let $\mathbf{C} = [\mathbf{c}_0, \mathbf{c}_1, \ldots, \mathbf{c}_{L-1}]$ be the discrete codec sequence. Fix a window length $P$ and stride $S \leq P$. We work with a left–padded extension in which $\mathbf{c}_i$ for $i < 0$ equals a [PAD] code so that all early windows have full length. For $k = 0, 1, \ldots$, define

$$\underbrace{\mathbf{C}^{(k)}}_{\text{context aggregation window}} \overset{\text{def}}{=} \mathbf{C}_{(k+1)S-P:(k+1)S} \qquad \text{and} \qquad \underbrace{\dot{\mathbf{C}}^{(k)}}_{\text{next span to predict}} \overset{\text{def}}{=} \mathbf{C}_{kS:(k+1)S},$$

so the stream is viewed as a stack of (potentially) overlapping windows $\{\mathbf{C}^{(k)}\}$ with step $S$, while the target of step $k$ is the length-$S$ slice $\dot{\mathbf{C}}^{(k)}$. Equivalently, the windowed view can be written as

$$\mathbf{C}[P; S] = \left[ \mathbf{C}_{S-P:S}, \ \mathbf{C}_{2S-P:2S}, \ \ldots, \ \mathbf{C}_{(L-P):L} \right].$$

**Decomposed generator.** Parameters are decomposed as $\theta = (\theta_{\text{AR}}, \theta_{\text{MD}})$. The autoregressive summarizer $\theta_{\text{AR}}$ produces a conditioning state from the accumulated history and text:

$$p_{\theta_{\text{AR}}}\left(\mathbf{h}_k \,\Big|\, \mathbf{C}^{(0)}, \mathbf{C}^{(1)}, \ldots, \mathbf{C}^{(k-1)}, \mathbf{X}\right),$$

capturing long-range dependencies across previously completed windows. Given $\mathbf{h}_k$, a bidirectional self-attention Transformer $\theta_{\text{MD}}$ then models the next-span distribution $p_{\theta_{\text{MD}}}\left(\dot{\mathbf{C}}^{(k)} \,\Big|\, \mathbf{h}_k\right)$, which will be realized via a masked diffusion objective over discrete tokens.

**Discrete masked diffusion over the next span.** Following the LLaDA-style formulation for discrete spaces, we specify a forward masking process and a reverse reconstruction process on $\dot{\mathbf{C}}^{(k)}$. Let $t \in [0, 1]$ and let $\lambda : [0, 1] \to [0, 1]$ be a monotonically increasing mask schedule. The forward process produces a partially masked sequence $\dot{\mathbf{C}}_t^{(k)}$ by independently replacing each position with a special token MASK with probability $\lambda(t)$ (and leaving it unchanged with probability $1 - \lambda(t)$); thus $\dot{\mathbf{C}}_1^{(k)}$ is fully masked and $\dot{\mathbf{C}}_0^{(k)}$ is clean. The reverse process is parameterized by a bidirectional Transformer that, at any $t$, predicts all masked sites simultaneously, i.e. $p_{\theta_{\text{MD}}}(\cdot \mid \dot{\mathbf{C}}_t^{(k)}, \mathbf{h}_k)$. Training

minimizes the cross-entropy only on the positions that are masked at time $t$, with a time-dependent weight that recovers an upper bound on the sequence negative log-likelihood:

$$\mathcal{L}(\theta_{\mathrm{MD}}) \overset{\text{def}}{=} - \mathbb{E}_{t, \, \dot{\mathbf{C}}_0^{(k)}, \, \dot{\mathbf{C}}_t^{(k)}} \left[ \frac{1}{t} \sum_j \mathbf{1}\left\{ \dot{\mathbf{C}}_{t,j}^{(k)} = \mathrm{MASK} \right\} \cdot \log p_{\theta_{\mathrm{MD}}}\left( \dot{\mathbf{C}}_{0,j}^{(k)} \, \middle| \, \dot{\mathbf{C}}_t^{(k)}, \mathbf{h}_k \right) \right]. \quad (2)$$

Here $j$ indexes token positions within the span, and $\mathbf{1}\{\cdot\}$ is the indicator. Conceptually, the AR module supplies a long-context sketch $\mathbf{h}_k$, while the masked diffusion infiller completes the $k$-th span in parallel, yielding a patchwise AR process with reduced exposure bias and high intra-patch fidelity.

At inference, generation proceeds through iterative decoding in parallel over a masked sequence. We begin from an all-masked initialization $\dot{\mathbf{C}}_1^{(k)}$. At time $t$, the masked diffusion Transformer takes $\dot{\mathbf{C}}_t^{(k)}$ as input and outputs categorical distributions for every currently masked position in one shot. From these distributions we produce a provisional completion $\widehat{\mathbf{C}}_t^{(k)}$ (sampling or choosing modes), and compute per-position confidence scores $s_t(i)$ (e.g., maximum class probability).

Let $N$ denote the total number of the decoding steps and define a monotonically decreasing mask budget $\rho_n = \lambda\left(1 - \frac{n}{N}\right)$ for each step $n$, we reapply masks to the least confident fraction $\rho_n$ of position, yielding the next iterate

$$\dot{\mathbf{C}}_{\rho_{n+1}}^{(k)} = \mathrm{Mask}\left( \widehat{\mathbf{C}}_{\rho_n}^{(k)}, \underset{\lfloor \rho_n |\Omega| \rfloor}{\arg\mathrm{top}}\left( - s_{\rho_n}(i) \right) \right),$$

where $\Omega$ indexes the currently editable sites, $s_t(i)$ denotes per-position confidence scores, $\widehat{\mathbf{C}}_{\rho_n}^{(k)}$ is produced as a provisional completion at this timestep, and $\mathrm{Mask}(\cdot, \cdot)$ replaces the selected entries with the mask token. This schedule enforces a monotone decrease in the masked ratio and mirrors the forward-noising trajectory, aligning the reverse refinement with the corruption process. Throughout decoding we employ classifier-free guidance (Ho & Salimans, 2022; Lin et al., 2024) with rescaling, following (Wang et al., 2024).

### 3.1.2 OVERALL ARCHITECTURE

Figure 1 sketches the DISTAR pipeline. In the spirit of DiTAR, we adopt a blockwise decomposition of the RVQ code stream: a long sequence of discrete tokens is sliced into patch-level units. Dependencies across patches are modeled by a causal autoregressive (AR) language model, whereas a Transformer-based masked diffusion module decodes within-patch content in parallel.

The text prompt is fed to the AR LM and then concatenated with a learned patch embedding obtained by aggregating the relevant RVQ codes. The LM produces a contextual summary $h$, which—together with a finite history of previously generated tokens—serves as conditioning for the diffusion model. Training minimizes the cross-entropy objective in equation 2.

DISTAR dispenses with both an explicit duration predictor and forced alignment. Moreover, unlike prior designs, the RVQ aggregator is not restricted to disjoint patches: overlapping windows are permitted. At each decoding step, the diffusion head predicts, in one shot, a segment whose length matches the aggregator's stride on the output stream, ensuring consistency. Additional architectural and training details are provided in the following sections.

### 3.2 AGGREGATOR

We use a lightweight non-autoregressive Transformer encoder to turn frame-level RVQ codes into one vector per patch. Concretely, each RVQ layer has its own embedding table, and we adopt factorized embedding parameterization (Lan et al., 2019): a narrow embedding (e.g., 32-d) is learned and then lifted to $d_{\mathrm{model}}$ by a layer-specific linear map. A learnable scalar mixes the layer embeddings at each frame into a single continuous vector by weighted summing, giving one hidden vector per RVQ frame.

Inspired by the overlapped mode in classic CNNs (O'shea & Nash, 2015), we do not require the stride to equal the aggregation patch size on the input RVQ sequence, thereby allowing overlapping.

With patch length $P$ and stride $S \leq P$, we intentionally allow $S < P$ so adjacent patches overlap, which smooths boundaries and provide more information. After the encoder, each patch vector is obtained by averaging the final hidden states over the last $S$ tokens of that patch. The resulting sequence of patch embeddings is then passed to the AR language model.

### 3.3 NEXT-PATCH MASKED DIFFUSION MODELING

To complete each drafted patch, we employ a discrete masked diffusion model that operates in parallel over the tokens of the patch while respecting bidirectional context, akin to LLaDA-style non-causal Transformers. Conditioning follows the spirit of (Jia et al., 2025) and (Wang et al., 2024): the diffusion model receives a compact prefix formed by concatenating (i) the autoregressive planner's hidden state and (ii) a sliding window of previously generated codes. To avoid scale mismatch, the AR hidden is first passed through a trainable scalar gate before concatenation, which keeps its contribution numerically stable.

The target RVQ streams are linearized across time into a one-dimensional token sequence $\tilde{\mathbf{c}} = (\mathbf{c}_k^\top, \mathbf{c}_{k+1}^\top, \ldots)$ with $\mathbf{c}_k$ denoting the code tuple at frame $k$. For each position, we add a learnable embedding and an RVQ-layer embedding to inject positional/type cues, which breaks symmetry and provides simple priors for the model (Dosovitskiy et al., 2020).

During training, we draw a timestep $t \sim \mathcal{U}(0, 1]$ and compute the masking ratio via a cosine schedule $\lambda(t) = \cos\left(\frac{1-t}{2}\pi\right)$ following (Chang et al., 2022). A fraction $\lambda(t)$ of target tokens in the current patch is replaced by a special [MASK] symbol, and the diffusion model learns to reconstruct all masked RVQ tokens simultaneously given the visible tokens and the conditioning prefix. At inference, we use the same cosine schedule to anneal the mask ratio over $N$ iterations using iteratively decoding, after which the process advances to the next patch.

### 3.4 TRAINING AND INFERENCE

**Embedding initialization.** For each discrete RVQ token, we bootstrap its embedding vector by *transplanting* the first 16 channels from the corresponding codebook of the RVQ codec. The remaining $d - 16$ channels are sampled i.i.d. from a Gaussian whose mean and variance are matched to those 16 copied channels. This preserves the native scale of the codebook while avoiding cold-start mismatch in the unused dimensions.

**Stochastic layer truncation.** To make the model resilient to depth reductions, during training we randomly drop the top $\ell$ RVQ tiers, drawing $\ell \sim \text{Unif}\{0, \ldots, L-1\}$ where $L$ is the total number of layers. The network therefore encounters examples with missing high-depth codes and learns to decode from shallower stacks, enabling test-time bitrate/compute control by simply pruning the last $\ell$ layers, with no retraining required.

**Classifier-free conditioning.** We implement classifier-free guidance (CFG) for the masked-diffusion module by independently dropping (i) the AR LM conditioning output and (ii) the past-code window with probabilities $0.1$ and $0.1$, respectively. At inference, the LM is evaluated once to produce its context, while the diffusion head executes multiple times. Unless stated otherwise, CFG is applied only to the historical code with a guidance scale of $1.25$ and a rescale factor of $0.75$.

**Decoding heuristics.** At inference, we observe a *tail-first* bias: tokens near the end of each patch often receive higher confidence early in decoding. Vanilla decoding thus induces a mask pattern misaligned with training and degrades performance. A likely reason is that, in temporally/casually dependent sequences, non-autoregressive training makes later positions easier (they lean more on preceding context), leading to overconfidence.

To mitigate this with three lightweight decoding tricks: (i) **Layer-wise temperature shaping.** Cool down deeper RVQ layers so they don't dominate early. Concretely, multiply the sampling temperature for layer $j$ by $T_{\text{layer}}^j$. (ii) **Position-wise temperature shaping.** Within a patch, reduce confidence for farther-ahead positions by scaling the temperature at offset $l$ with $T_{\text{time}}^l$ ($0 \leq l < S$). (iii) **Hybrid sampling.** Generate the first $50\%$ of the target positions by sampling, then switch to greedy for

the remaining 50% to trade diversity for stability. In principle, the greedy/sample schedule is a hyperparameter; we adopt a simple half–half scheme to avoid over-tuning.

In our default setup, $T_{\text{layer}}$=0.8, $T_{\text{time}}$=0.95; we use top-$k$=50, top-$p$=0.9, and anneal temperature from 1.0 to 0.1 for sampling. Following (Chen et al., 2024), we also add a repetition-aware penalty within every $P_r$=4 patches to curb code collapse at the layer level.

Together these inference strategies keep sampling diverse yet stable, while also enabling high-quality greedy decoding.

### 3.5 IMPLEMENTATION DETAILS

#### 3.5.1 RESIDUAL VECTOR QUANTIZER

We largely adopt the overall architecture and staged training recipe of MAGICODEC (Song et al., 2025a), but replace its single-codebook VQ module with a residual vector quantizer (RVQ). Our RVQ is a Transformer-based streaming codec with approximately 0.3B parameters. Under this configuration, a 24 kHz waveform is compressed into a discrete token stream at 64 Hz. The codec employs 9 residual stages (RVQ layers); each stage uses a codebook of size 65,536 with 16-dimensional code vectors.

#### 3.5.2 MODEL

DISTAR comprises (i) an aggregator, (ii) a causal language model (LM), and (iii) a masked diffusion model (MDM), all instantiated with Transformer backbones. The aggregator and MDM are bidirectional RoFormers with rotary positional encodings (RoPE) (Su et al., 2024), SwiGLU activations (Shazeer, 2020), and a Pre-Norm layout using RMSNorm(Zhang & Sennrich, 2019). The AR LM follows the Qwen2.5 (Yang et al., 2024) style decoder-only architecture and is trained from scratch. Input text is converted to phoneme sequences and embedded via a learned lookup table. At inference, consistent with prior practice, we construct a prefix context that includes the text together with a short acoustic prompt.

#### 3.5.3 OPTIMIZATION

Training uses the cut cross entropy (Wijmans et al., 2024) as the implementation of cross entropy for masked token prediction. We implement SwiGLU, RMSNorm, and RoPE via Liger's open-source Triton kernels (Hsu et al., 2024), and optimize with a fused Adam variant. Further details are provided in the Appendix B.1.

## 4 EXPERIMENTS

In this subsection, we position DISTAR against strong contemporary systems and report state-of-the-art results.

### 4.1 EXPERIMENTAL SETTINGS

**Datasets.** All models are trained on Emilia (He et al., 2024), a large, multilingual, in-the-wild speech corpus curated for scalable speech generation. For this study we use only its English subset, totaling roughly 50k hours. Evaluation spans two open benchmarks: (i) LibriSpeech(PC) (Meister et al., 2023) test-clean, 5.4 hours from 40 unique English speakers; we follow the established zero-shot cross-sentence protocol and adopt the subset from F5TTS (Chen et al., 2025b) (40 prompts, 1127 utterances), and (ii) SeedTTS test-en, introduced with Seed-TTS (Anastassiou et al., 2024), consisting of 1088 English samples drawn from Common Voice (Ardila et al., 2019).

**Evaluation Metrics.** We use both objective and subjective metrics to evaluate our models. For the objective metrics, we evaluate (i) Word Error Rate (WER) to assess robustness and intelligibility using Whisper-large-v3 (Radford et al., 2022) as our ASR model; (ii) Speaker similarity (SIM) via cosine similarity between the TDNN-based WavLM embeddings (Chen et al., 2022) extracted from the generated audio and its reference prompt; (iii) UTMOS (Saeki et al., 2022), an automatic

Table 1: Evaluation results of DISTAR and other systems LibriSpeech-PC test-clean and SeedTTS test-en. ♦ denotes the scores reported in DiTAR paper. The boldface and underline indicate the best and the second-best result, respectively. ↑ and ↓ indicate that lower or higher values are better.

| System | #Params | WER(%)↓ | SIM↑ | UTMOS↑ |
|---|---|---|---|---|
| **LibriSpeech test-clean** | | | | |
| Human | - | 1.80 | 0.69 | 4.10 |
| RVQ resynthesized | - | 1.83 | 0.66 | 4.23 |
| IndexTTS | 0.5B | 2.57 | 0.62 | **4.35** |
| E2TTS (NFE=32) | 0.3B | 2.74 | **0.70** | 3.47 |
| F5TTS-v1 (NFE=32) | 0.3B | 2.02 | 0.68 | 3.83 |
| DiTAR (NFE=10) ♦ | 0.6B | 2.39 | 0.67 | 4.22 |
| DiSTAR-base (NFE=24) | 0.15B | 1.90 | 0.64 | 4.29 |
| DiSTAR-medium (NFE=24) | 0.3B | **1.66** | 0.67 | 4.27 |
| **Seed-TTS test-en** | | | | |
| Human | - | 1.47 | 0.73 | 3.53 |
| RVQ resynthesized | - | 1.71 | 0.70 | 3.79 |
| IndexTTS | 0.5B | 1.92 | 0.61 | 3.98 |
| E2TTS (NFE=32) | 0.3B | 2.20 | **0.71** | 3.20 |
| F5TTS-v1 (NFE=32) | 0.3B | 1.35 | 0.68 | 3.66 |
| DiTAR (NFE=10) ♦ | 0.6B | 1.78 | 0.64 | **4.15** |
| DiSTAR-base (NFE=24) | 0.15B | 1.51 | 0.64 | 3.93 |
| DiSTAR-medium (NFE=24) | 0.3B | **1.32** | 0.66 | 4.05 |

predicted mean opinion score (MOS) for speech quality.For the subjective metrics, comparative mean option score (CMOS) and similarity mean option score (SMOS) are used to evaluate naturalness/robustness and similarity, respectively. CMOS is on a scale of -3 to 3, and SMOS is on a scale of 1 to 5.

**Model settings and baselines.** We train all models on 64 NVIDIA A100 80GB GPUs. We train DISTAR of two sizes DISTAR-base (0.15B), and DISTAR-medium (0.3B). Architectural specifics are provided in Appendix B.2. Unless otherwise noted, we use a patch size of 8 and condition the diffusion module on a single historical patch. Inference procedures are detailed in Section 3.4.

## 4.2 ZERO-SHOT TTS

we show comparison results with SOTA baselines. The main results of objective metrics are shown in Table 1. Subjective results are detailed in Table 2.

DISTAR exhibits strong overall performance. Across benchmarks it attains the lowest WER, indicating robust synthesis. We assess speaker similarity using both objective and human judgments: SIM and speaker SMOS. DISTAR yields SIM on par with the best alternatives and leads on SMOS. We attribute these gains in part to reduced sensitivity to high-frequency artifacts in the reference prompt, which preserves cleaner timbral cues during cloning. The same trend is observed in the objective predictor UTMOS and in CMOS listening tests. Notably, relative to continuous-representation systems, DISTAR achieves comparable or superior perceptual quality with a similar or smaller parameter budget. As model capacity grows, DISTAR yields consistent improvements on objective metrics, closely matching the scaling behavior reported for discrete-token autoregressive systems and indicating a healthy scaling trajectory.

## 4.3 ABLATION STUDY

In this subsection, we conduct a detailed analysis of the various components of DISTAR. Unless specified otherwise, we default to using DISTAR-base with 0.15 billion parameters, a patch size of 8, a stride of 8, and NFE of 24, tested on the LibriSpeech-PC test dataset mentioned above. We discuss the patch size in Appendix D and classifier-guidance free settings in Appendix C.

**Decoding Strategies.** We compare DISTAR under multiple inference strategies (Table 3). Deterministic greedy decoding yields the lowest WER, evidencing stable synthesis, while speaker

Table 2: Subjective evaluation results on Seed-TTS test-en dataset. We compare DiTAR with several leading TTS systems based on discrete or/and continuous speech representations.

| System | SMOS | CMOS |
|---|---|---|
| Human | 3.07 | 0.00 |
| FireRedTTS | $2.36_{\pm 0.58}$ | $-0.34_{\pm 0.21}$ |
| CosyVoice 2 | $3.07_{\pm 0.21}$ | $-0.04_{\pm 0.17}$ |
| E2TTS | $3.29_{\pm 0.19}$ | $-0.08_{\pm 0.22}$ |
| F5TTS | $3.08_{\pm 0.20}$ | $0.01_{\pm 0.12}$ |
| DiSTAR | $\mathbf{3.31}_{\pm 0.25}$ | $\mathbf{0.22}_{\pm 0.13}$ |

Table 3: Objective evaluation results between different decoding strategies.

| Type | $T_{\text{time}}$ | $T_{\text{layer}}$ | WER | SPK |
|---|---|---|---|---|
| Sample | 1 | 1 | 2.11 | 0.626 |
| Sample | 0.95 | 0.8 | 1.99 | **0.640** |
| Greedy | 0.95 | 0.8 | **1.91** | 0.636 |

similarity is slightly lower than with stochastic sampling. This aligns with the standard diversity-determinism trade-off: sampling can recover timbral nuances at the cost of higher variability.

## 4.4 INFERENCE EFFICIENCY AND CONTROLLABILITY

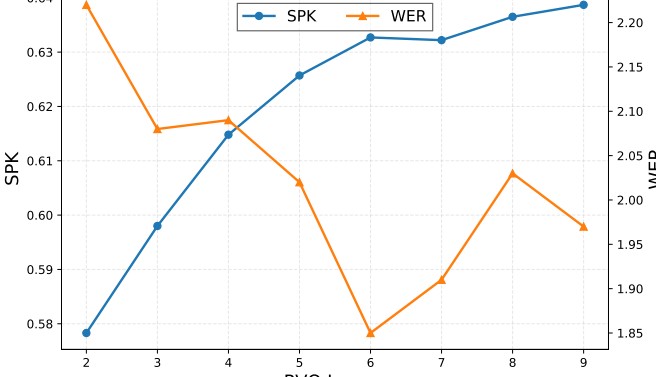

Figure 2: Comparison results among different inference RVQ layers.

During training, DISTAR randomly drops the last $\ell$ RVQ layers. At inference, we simply prune higher RVQ layers to achieve variable bitrate and compute. Figure 2 summarizes the inference–quality trade-off under different RVQ pruning depths. As more RVQ layers are retained (hence higher FLOPs), speaker similarity increases markedly, whereas WER changes little and reaches its minimum around six layers. This pattern is consistent with the hypothesis that upper RVQ layers primarily encode acoustic detail rather than linguistic content (Chen et al., 2025a).

## 5 CONCLUSION

We presented DISTAR, a zero-shot TTS framework that operates in an RVQ code space and tightly couples an autoregressive Transformer with masked diffusion. DISTAR achieves blockwise parallelism while effectively modeling layer–time dependencies across RVQ levels and local temporal neighborhoods, without an explicit duration predictor. We further introduced a simple but effective RVQ-aware sampling procedure that stabilizes inference and improves perceptual quality. In combination, these design choices yield SOTA robustness, speaker similarity, and naturalness in zero-shot speech synthesis, while exposing clear levers for controllability at inference time.

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

## A  LIMITATIONS AND SCOPE

Our results are currently demonstrated under the codec, RVQ configuration, and evaluation setups described in the paper. Performance may depend on the RVQ depth and codebook design. Due to resource constraints, we trained our model solely on a around-50k-hour English corpusHe et al. (2024); broader generalization to multilingual and multi-style settings remains to be evaluated.

## B  Implementation Details

### B.1  Training Details

We utilize 64 A100 GPUs, each processing a batch size of 36K token frames, and train DISTAR for 0.6M steps. The AdamW optimizer is employed with a learning rate of 0.75e-4 for AR, and 1.5e-4 for the other parts, $\beta_t = 0.9$, and $\beta_2 = 0.99$.

For masked token prediction we adopt Cut Cross-Entropy (CCE)[1] (Wijmans et al., 2024), which computes the cross-entropy loss without materializing the full $|V|$-way logit tensor in GPU global memory. We use Liger [2] (Hsu et al., 2024) kernels, which are drop-in replacements that fuse fine-grained operations and use recomputation, in-place updates, and program coarsening to cut high-bandwidth-memory traffic and kernel launches.

### B.2  Model Architecture

Table 4: Configurations of DiSTAR with different sizes.

| Model size | | $\sim 0.15\text{B}$ | $\sim 0.3\text{B}$ |
|---|---|---|---|
| Aggregator | RVQ dim | 32 | 48 |
| | Number of layers | 4 | 4 |
| | Hidden dim | 512 | 768 |
| | Number of heads | 8 | 16 |
| | FFN dim | 1024 | 3072 |
| Language Model | Number of layers | 24 | 24 |
| | Hidden dim | 512 | 768 |
| | Number of heads | 8 | 16 |
| | FFN dim | 1024 | 2048 |
| Diffusion | Number of layers | 16 | 16 |
| | Hidden dim | 512 | 768 |
| | Number of heads | 16 | 16 |
| | FFN dim | 2048 | 3072 |

Table 4 presents the key hyperparameters of the models.

## C  Classifier-Free Guidance

Classifier-free guidance (Ho & Salimans, 2022) is widely used to strengthen conditional adherence in diffusion models by jointly training conditional and unconditional modes within a single network via random condition dropping. We implement CFG for the masked-diffusion module (MDM). Let $g_\theta(\mathbf{C} \mid \cdot)$ denote the last-layer embedding (pre-projection) for masked positions, where $\mathbf{C}$ are target codes, $\mathbf{C}^p$ is the history code context, and $\mathbf{h}$ is the AR condition. During training, we randomly drop the AR condition and the historical prompt with probabilities 0.1 and 0.1, respectively, to expose $p_{\theta_{\text{MD}}}$ to unconditional variants.

Define the three evaluations:

$$g_{\text{hist}} = g_\theta(\mathbf{C} \mid \mathbf{C}^p, \mathbf{h}), \quad g_{\text{ar}} = g_\theta(\mathbf{C} \mid \mathbf{C}^p), \quad g_{\text{uncond}} = g_\theta(\mathbf{C}).$$

We consider two CFG constructions for the final-layer embedding:

**(A) History-only CFG.**
$$\tilde{g}^{(A)} = g_{\text{hist}} + w_{\text{cfg,hist}}\big(g_{\text{hist}} - g_{\text{ar}}\big). \tag{3}$$

**(B) Nested AR+history CFG.** First form an AR-guided intermediate,
$$\hat{g}_{\text{ar}} = g_{\text{ar}} + w_{\text{cfg,ar}}\big(g_{\text{ar}} - g_{\text{uncond}}\big), \tag{4}$$

---

[1] https://github.com/apple/ml-cross-entropy
[2] https://github.com/linkedin/Liger-Kernel

then apply history guidance on top:

$$\tilde{g}^{(B)} = g_{\text{hist}} + w_{\text{cfg,hist}}\big(g_{\text{hist}} - \hat{g}_{\text{ar}}\big). \tag{5}$$

To tame over-exposure at high $w_{\text{cfg},\cdot}$, we apply the variance-matching rescale of (Lin et al., 2024):

$$g^{\text{rescale}} = \tilde{g} \cdot \frac{\text{std}\big(g_{\text{hist}}\big)}{\text{std}\big(\tilde{g}\big) + \varepsilon}, \quad \varepsilon > 0, \tag{6}$$

where $\text{std}(\cdot)$ is computed over the embedding dimension.

Finally, we optionally blend the rescaled and un-rescaled outputs:

$$g^{\text{final}} = w_{\text{rescale}}\, g^{\text{rescale}} + \big(1 - w_{\text{rescale}}\big)\, \tilde{g}, \quad w_{\text{rescale}} \in [0, 1]. \tag{7}$$

In our ablations, we instantiate $\tilde{g}$ with either Eq. equation 3 or Eq. equation 5.

As reported in Table 5, holding all other factors fixed, nested CFG (Scheme **B**) and simple single-step CFG (Scheme **A**) yield nearly identical objective metrics. To reduce computational cost and latency, we therefore adopt the simpler Scheme **A** (single-step CFG) as the default in the main experiments.

Table 5: Objective evaluation results between different classifer-free guidance strategies.

| Schema | $w_{\text{cfg,hist}}$ | $w_{\text{cfg,ar}}$ | $w_{\text{rescale}}$ | WER | SPK |
|--------|------|------|------|------|------|
| **A** | 2 | - | 0.75 | 2.12 | **0.63** |
| **A** | 1.25 | - | 0.75 | **1.74** | **0.63** |
| **A** | 0.75 | - | 0.75 | 1.99 | 0.62 |
| **B** | 0.75 | 0.75 | 0.75 | 1.76 | **0.63** |
| **B** | 0.75 | 1.25 | 0.75 | 1.90 | **0.63** |
| **B** | 1.25 | 1.25 | 0.75 | 1.88 | **0.63** |

## D PATCH SIZE.

Table 6: Objective evaluation results between different patch size $P$ of DiSTAR. All results use greedy decoding to eliminate sampling randomness.

| Patch size | WER | SPK | UTMOS |
|--------|------|------|------|
| 2 | 4.50 | 0.63 | 4.26 |
| 4 | **1.85** | **0.65** | **4.33** |
| 8 | 1.91 | 0.64 | 4.29 |

To isolate the effect of patching, we vary the patch size $P$ while keeping the DISTAR-base configuration fixed and adopt greedy decoding to eliminate sampling stochasticity. As shown in Table 6, performance degrades when $P$ is either too small or too large. Small patches deprive the masked diffusion of sufficient within-patch context, weakening reconstruction; moreover, the autoregressive horizon is only marginally shortened, so efficiency gains are limited. In contrast, very large patches encourage the refiner to overrely on copy-from-context shortcuts rather than resolving long–time dependencies, which also harms quality. Balancing these factors, we use $P = 8$ by default as a favorable trade-off between compute and performance.

## E EVALUATION BASELINES

**DiTAR (Jia et al., 2025).** A *patch-based* autoregressive framework on continuous tokens that couples a causal LM with a bidirectional diffusion transformer (LocDiT) for intra-patch prediction. This divide-and-conquer design balances determinism and diversity and reduces compute versus fully diffusion-based approaches. The paper reports strong zero-shot TTS with fast temperature-based sampling.

**CosyVoice2 (Du et al., 2024b).** A scalable zero-shot TTS that unifies offline and streaming synthesis in one framework via a unified text–speech LM and a chunk-aware causal flow-matching decoder. It replaces VQ with finite-scalar quantization (FSQ) to improve codebook utilization and simplifies the LM by removing explicit speaker embeddings while initializing from a pretrained LLM. CosyVoice2 supports rich instruction following (emotion, accent, role) and reports near-lossless quality in streaming mode compared with offline synthesis. We use the official code and pre-trained checkpoint [3].

**E2TTS (Eskimez et al., 2024).** A fully non-autoregressive, flow-matching TTS that treats generation as an audio-infilling task: text is padded with filler tokens and a mel-spectrogram generator is trained via flow matching. Its simplicity established a modern NAR baseline later built upon by follow-ups (Eskimez et al., 2024). We use the unofficial implementation and pre-trained checkpoint [4].

**F5TTS (Chen et al., 2025b).** A diffusion-transformer (DiT) flow-matching system that addresses E2TTS's convergence and robustness issues with a ConvNeXt (Liu et al., 2022) text encoder and an inference-time Sway Sampling schedule. Trained at scale (about 100k hours), it demonstrates strong zero-shot cloning, code-switching, and efficient inference, with open-sourced code and checkpoints. It is widely adopted as a strong continuous-latent NAR baseline. We use the official code and pre-trained checkpoint [5].

**IndexTTS (Deng et al., 2025).** An industrial-level, controllable zero-shot TTS that builds on XTTS (Casanova et al., 2024)-style AR modeling with practical enhancements for stability and control. The system targets faster inference and simpler training while providing fine control over pronunciation, timing, and emotion, with public demos and code. We use the official code and pre-trained checkpoint [6].

**FireRedTTS (Guo et al., 2024).** A foundation TTS framework with a semantic-token LM frontend and a two-stage waveform generator, designed for robust zero-shot cloning and instruction-tuned conversational speech. It emphasizes data/process pipeline design for large-scale training and showcases in-context learning for dubbing and chat. We use the official code and pre-trained checkpoint [7].

# F USE OF LARGE LANGUAGE MODELS.

An LLM was used to assist with language polishing and clarity only. All ideas, experimental designs, analyses, and conclusions are our own.

# G BOARDER IMPACT

High-fidelity zero-shot TTS offers clear benefits for accessibility, education, and creative production, yet the ability of DISTAR to closely match speaker timbre introduces material risks, such as impersonation and social-engineering fraud, spoofed voice biometrics, non-consensual cloning, and scalable disinformation. To mitigate misuse, we advocate consent-first deployment with strict use-policy gating, the use of robust audio watermarks to support provenance, and a public abuse-reporting channel with prompt triage and access revocation.

---

[3] https://huggingface.co/FunAudioLLM/CosyVoice2-0.5B
[4] https://huggingface.co/SWivid/E2-TTS
[5] https://huggingface.co/SWivid/F5-TTS
[6] https://huggingface.co/IndexTeam/Index-TTS
[7] https://huggingface.co/FireRedTeam/FireRedTTS

