# OpenReview forum: "DiSTAR: Diffusion over a Scalable Token Autoregressive Representation for Speech Generation"
_ICLR.cc/2026/Conference — Submitted to ICLR 2026_

### Official Review · Reviewer_rLfL · 2025-10-22

**Soundness:** 2
**Presentation:** 2
**Contribution:** 2
**Rating:** 2
**Confidence:** 4

**Summary:**

The paper extends DiTAR by replacing continuous code with RVQ codes and use a LLaDA style masked diffusion transformer to predict the next code patch.

**Strengths:**

1. There is some limited novelty in combining LLaDA style diffusion transformer with DiTAR approach.

**Weaknesses:**

1. The paper is a bit difficult to read with some grammar issues. If possible, I suggest the authors to seek help from native English speakers to make the paper more reader friendly.
2. The authors claim the model to be SOTA in robustness, speaker similarity and naturalness but the results in Table 1 seems to indicate otherwise? The Speaker SIM and UTMOS scores are lower than competitors.
3. The evaluation section seems a bit sketchy overall. Why are the models compared in Table 2 different from Table 1? Subjective evaluations are missing key details (e.g. number of evaluators, number of samples per evaluator). The ablation study only covers decoding strategies but not other design choices.

**Questions:**

See weakness.

---

### Official Review · Reviewer_tPVS · 2025-10-31

**Soundness:** 3
**Presentation:** 2
**Contribution:** 2
**Rating:** 4
**Confidence:** 4

**Summary:**

In this paper, authors proposes DiSTAR, a zero-shot TTS framework that works entirely in a discrete residual vector quantizatio code space, coupling an autoregressive language model (sketcher) with a masked diffusion Transformer (refiner). The approach avoids forced alignments and duration predictors, instead using blockwise parallelism where the AR model drafts RVQ token sketches for each patch and the diffusion model performs parallel masked infilling to complete the block. Using DiSTAR discrete latent space can be directly used for controllability, supports a variety of decoding strategies, and allows inference-time bitrate and compute control by pruning RVQ layers. The system is evaluated on standard zero-shot TTS benchmarks. DiSTAR  demonstrates improvements over recent baselines in robustness, naturalness, and speaker consistency.

**Strengths:**

(a) The paper presents a integration of an autoregressive LM and a diffusion model operating on RVQ discrete tokens. This combination addresses weaknesses of purely-AR or purely-diffusion approaches. Furthermore, the idea of iterative discrete demasking (inspired by LLaDA) is technically interesting and new in the TTS domain.

(b) The empirical results back up the claims of improved robustness, high naturalness, and better speaker consistency (SMOS) across unseen voices.

(c) Eliminating the need for forced aligners, duration models, or external text-speech alignment is a another practical strength of the proposed work.

**Weaknesses:**

(a) The proposed appraoch is the combination of different techniques, each individual component draws on previously known ideas, so the perceived novelty is Incremental. DiSTAR’s core innovation is applying masked diffusion in the discrete RVQ domain, which is new, but conceptually it parallels prior AR+refinement pipelines and the LLaDA diffusion LM approach in NLP.

(b) The method is complex and lack in clarity. Furthermore the system involves multiple components and a non-trivial training procedure, which are not fully transparent in the description .For example i cannot understand clearly how the AR hidden sketch is defined and used.  Is it generating one coarse codebook stream, a fused embedding per frame, or something else?

(c) The results claim comparable inference speed to a baseline (DiTAR), but since it still relies on an iterative diffusion process for each patch, which may be a bottleneck.

(d) Couple of relevant baselines are absent. In particular, there is no direct comparison to a pure AR discrete token model of comparable size on the same data. Without an explicit AR-only baseline, it’s hard to isolate how much the diffusion refiner helps beyond a standard AR approach

(e) I think an ablation where the diffusion module is removed (i.e. the AR alone generates all codebooks) would be insightful. Does the diffusion mainly help with fine detail, or also with stability (WER)?

(f) The authors mention that DiSTAR is less sensitive to high-frequency artifacts in the reference prompt than others, attributing better speaker cloning to this. However, it’s unclear why ? Is the diffusion refiner helps ignore prompt noise.? There is a need to evaluate the robustness under prompt domain shift.

**Questions:**

(a) How are the AR drafter and diffusion refiner trained jointly or sequentially? It is implied in the paper that a shared token space allowing end-to-end optimization , but can you clarify if you train the AR LM and the diffusion Transformer simultaneously or in stages ?

(b) Did you test scenarios beyond the lengths in the benchmark (e.g., generating several minutes of speech concatenating multiple paragraphs)? Does the model maintain speaker identity and prosody consistently in truly long sequences?

(c) Could you provide more details on pruning? For example, if you drop the top $k$ codebooks at inference (using only the first $L-k$ RVQ layers), how does it impact MOS or WER?

---

### Official Review · Reviewer_mwhZ · 2025-11-01

**Soundness:** 2
**Presentation:** 3
**Contribution:** 3
**Rating:** 4
**Confidence:** 3

**Summary:**

DiSTAR is a method involving the recently popular paradigm of combining the benefits of autoregressive decoder-only LMs with diffusion models. In this specific instance, the entire architecture relies only on discrete tokens from an RVQ codec-based audio tokenizer, which is unlike previous work (DiTAR) where continuous latents are used. Consequently, the diffusion process is now a masked diffusion model.

The DiSTAR architecture involves aggregated patch-wise tokens fed to the AR model which “sketches” the next patch. The MDM refines the aggregate token into the RVQ tokens conditioned on previous token predictions. The method involves training tricks like dropping out RVQ layers that help the model remain robust across a wide range of bitrates.

Comparing with other state-of-the-art TTS models shows that the DiSTAR achieves comparable quality across various metrics.

**Strengths:**

The main strength of the paper is the fact that it shows how to apply the AR + Diffusion paradigm to TTS using multi-level RVQ discrete audio tokens which helps remove the need for separate duration predictors and stop predictor; simply predicting [eos] tokens in the AR step is enough. This approach of patch-wise AR prediction mitigates some of the error-accumulation issues since the finer RVQ tokens are being generated in the masked diffusion sampling stage. The results also look good, with both subjective and objective metrics showing comparable results against strong baselines.

**Weaknesses:**

Weaknesses and questions:
- The authors use some embedding initialization trick but do not cite any existing work or ablate the design to prove it is effective.
- Similarly the utility of stochastic layer truncation is not cited/ablated. I believe the DAC (descript audio codec) paper does use this technique in the training of DAC but the authors are using it in the training of the LM and MDM on top of the RVQ codec. Will this still be needed if the authors used DAC or the RVQ decoder is already trained with quantizer dropout?
- The authors mention in the abstract that AR+Diffusion models on continuous latents are brittle under distribution shifts but do not really run any experiments that compare DiTAR vs DiSTAR under such a setting.
- The claims in the abstract regarding surpassing state-of-the-art for speaker similarity seem a little exaggerated based on the results shown in the paper. SMOS is very close to E2TTS with a wide spread, and the objective SIM metric is also lower than some other baselines.

**Questions:**

Please see the weakness section.

---

### Meta-Review · Area_Chair_atXA · 2026-01-05

**Summary:**

This paper presents a method called DiSTAR which is a diffusion method for text-to-speech.  The reviewers unfortunately all recommended rejection, with two borderline rejects and one reject (4, 4, 2).  The reviewers all complained about clarity issues.  They found that the method seemed like a complex combination of (incremental) contributions but lacked ablations to understand where gains were coming from.  One reviewer complained that the paper contained a number of 'tricks' or heuristics without explanation, justification or ablations.  Multiple reviewers found the evaluation to be somewhat questionable ("a bit sketchy") and didn't really seem to justify the claims made in the paper.

The authors don't seem to have provided a rebuttal.  Therefore the recommendation is to reject the paper.  Hopefully the reviews will be helpful for the authors to understand how to improve the work and the manuscript for a future submission.

**Reviewer Concerns:**

The authors didn't provide a response.

**Reviewer Scores:**

The authors didn't provide a response and the reviews seem reasonable.

---

### Decision · Program_Chairs · 2026-01-26

Reject